# Nucleofection as an Efficient Method for Alpha TC1-6 Cell Line Transfection

**Marija Đorđević** [1], **Verica Paunović** [2], **Maja Jovanović Tucović** [3], **Anja Tolić** [1], **Jovana Rajić** [1], **Svetlana Dinić** [1], **Aleksandra Uskoković** [1], **Nevena Grdović** [1], **Mirjana Mihailović** [1], **Ivanka Marković** [3], **Jelena Arambašić Jovanović** [1,*] **and Melita Vidaković** [1]

1   Department of Molecular Biology, Institute for Biological Research "Siniša Stanković"—National Institute of Republic of Serbia, University of Belgrade, Bulevar Despota Stefana 142, 11060 Belgrade, Serbia

2   Institute of Microbiology and Immunology, Faculty of Medicine, University of Belgrade, Dr. Subotića 1, 11000 Belgrade, Serbia

3   Institute of Medical and Clinical Biochemistry, Faculty of Medicine, University of Belgrade, Pasterova 2, 11000 Belgrade, Serbia

*   Correspondence: jelena.arambasic@ibiss.bg.ac.rs; Tel.: +381-11-2078-343

**Abstract:** An efficient transfection is a crucial step for the introduction of epigenetic modification in host cells, and there is a need for an optimized transfection process for individual model systems separately. Mouse pancreatic αTC1-6 cells, which act as an attractive model system for epigenetic cell reprogramming and diabetes treatment, were transiently transfected with two different transfection methods: the chemical method with polyethyleneimine (PEI) and nucleofection as a physical transfection method. Flow cytometry and fluorescent microscopy examination of GFP expression showed that transfection efficiency was affected by the size of plasmids using both transfection methods. Subsequently, the Cas9 mRNA expression confirmed successful transfection with EpiCRISPR plasmid, whereas the cell physiology remained unchanged. The adjusted nucleofection protocol for αTC1-6 cells transfected with an EpiCRISPR mix of plasmids reached 71.1% of GFP-positive transfected cells on the fifth post-transfection day and proved to be much more efficient than the 3.8% GFP-positive PEI transfected cells. Modifying the protocol, we finally specify CM-156 program and SF 4D-Nucleofector X Solutions for Amaxa™ nucleofection as a method of choice for alpha TC1-6 cell line transfection.

**Keywords:** alpha TC1-6 cells; transfection; PEI; nucleofection; green fluorescent protein; EpiCRISPR

## 1. Introduction

The modification of specific epigenetic signatures which define cell identity and plasticity could lead to the change of cell function without alteration in genomic DNA sequences. Powerful clustered regularly interspaced short palindromic repeats (CRISPR/Cas; CRISPR-associated protein) genome-targeting technology could be redesigned for epigenome editing and targeted modulation of the gene expression (EpiCRISPR). This EpiCRISPR system contains nuclease-deactivated Cas9 (dCas9) fused with an enzyme that can change certain marks on DNA or histone tails close to the targeted DNA sequence, offering advantages over the other epigenome-editing technologies such as the simpler design for targeting new sequences [1–3]. Studies have found that combinations of several epigenetic effector domains achieve stronger effects than using only one element [4].

The pancreatic alpha cells from Langerhans islets represent a source of the hormone glucagon which is responsible for elevating blood glucose levels and maintaining glucose homeostasis. Alpha cells have reached the spotlight of scientific discovery because they serve as a protector of beta cells supporting their insulin-producing capacity [5]. Because diabetes is caused by beta cell loss, great attention is focused on alpha cells as a promising

source for innovative diabetes therapies due to their ability to spontaneously transdifferentiate into insulin-producing cells [6]. Alpha cell hyperplasia in response to beta cell injuries offers promise as a source for increasing insulin-producing cells [7]. Because alpha cells are considered as a proper source for beta cell replacement, the development of new (epi)genetic manipulation strategies that will push alpha cells into the transdifferentiation process is needed [8].

The efficiency of introducing epigenetic modification and consequently potential biomedical application depend on the efficient delivery of foreign DNA into the host cells through the transfection process [9]. There is no ideal transfection method for different cell types originating from the same organism and especially from different sources such as plant, animal, and bacterial cells. All the transfection methods are roughly classified into viral- and non-viral-based. The non-viral transfection method can be further classified into chemical and physical/mechanical methods [10]. Each of these methods has its particular advantages and weaknesses. The method of choice depends on the experimental design and objective [9].

The chemical methods for transfection are the most widely used in research. These methods are based on the use of cationic polymer, calcium phosphate, cationic lipid, and cationic amino acid [9]. The most typical cationic polymer for delivering plasmid in vitro is polyethyleneimine (PEI). PEI efficiently induces electrostatic condensation of DNA molecules, forming positively charged nanometer-sized particles which interact with the negatively charged components of the cell membrane for endocytosis. Afterward, DNA is released into the cytoplasm [11]. PEI is broadly used because of its accessibility, high DNA delivery efficiency, and reproducibility during upscaling. DNA-PEI particle size and transfection efficiency could be affected by the many parameters, such as the amount of DNA, the DNA ratio to the PEI, the timing and the solution conditions for complex formation, the transfection medium, and cell density at the time of transfer [12].

The physical/mechanical transfection method includes electroporation, sonoporation, biolistic particle delivery, gene microinjection, and laser irradiation [13,14], with electroporation being the most commonly used. Electroporation introduces DNA into a variety of cells by using pulsed electrical fields. Several instruments for electroporation are commercially available, and manufacturers supply guidelines for optimization of electroporation and protocols for specific cell types. Amaxa™ 4D-Nucleofector™ (Lonza, Germany) is a modular and scalable unit for electroporation-based nucleofection and is used for delivering the widest range of substrates such as DNA, RNA, proteins, and small molecules directly to nuclei of targeted cells. Amaxa™ 4D-Nucleofector™ is indicated as an easy-to-use technology and offers several benefits over traditional electroporation methods, providing high transfection efficiency for difficult-to-transfect cells, low cytotoxicity, and preservation of functionality. A specific combination of optimized pulse and solution (transfection buffers) contributes to minimal cytotoxicity of the Nucleofector® Technology, thus preserving the functionality of the cells. Because 4D Nucleofector® Technology offers virus- and reagent-free transfection, there is no reagent toxicity or immunogenic effects [15,16]. This technology allows maintenance of phenotypic markers and differentiation potential, which is very important for research in hematopoietic and stem cells [17,18]. The main disadvantages of nucleofection are the requirement for expensive consumables and equipment, and cell mortality under suboptimal conditions. The components of buffers are not identified, and the parameters of programs cannot be controlled by the user.

Regardless of the transfection method, the crucial conditions for efficient reception of substrate are healthy and actively dividing cells [19]. Also, important parameters that could greatly influence transfection efficiency are the number of cell passages, degree of cell confluency, and contamination, as well as the quality, quantity, and size of DNA to be transfected. The efficiency of gene delivery depends on plasmid DNA size. Increasing plasmid size dramatically reduces the efficiency [20], probably due to a decreased delivery of larger DNA across the plasma/nuclear membrane [21].

The aim of this study was to determine the optimal transfection method for use in mouse pancreatic alpha TC1-6 cells. Cells were transfected with two different types of transfection methods: chemical transfection mediated with PEI, and nucleoporation with 4D-Nucleofector™ Technology. Cells were transfected with a plasmid-containing reporter gene for green fluorescent protein (GFP), providing information about transfection efficiency alone or in combination with more plasmid-caring information about EpiCRISPR. This study specifies that nucleoporation with 4D-Nucleofector™ is the preferred transfection method for in vitro transfection of mouse pancreatic alpha TC1-6 cell line. It may be useful in gene expression studies in the field of cellular reprogramming technology for pancreatic beta cell regeneration.

## 2. Materials and Methods

### 2.1. Cell Culture

A mouse pancreatic alpha TC1 clone 6 cell line ($\alpha$TC1-6, CRL-2934, American Type Culture Collection, Manassas, VA, USA) was cultured in 15 mM glucose Dulbecco's Modified Eagle's medium (DMEM), made by mixing high- (Gibco, by Thermo Fisher Scientific, Bremen, Germany) and low-glucose DMEM (Sigma-Aldrich, St. Louis, MO, USA) (1:1, *v:v*), and supplemented with 10% fetal bovine serum (FBS) (Sigma-Aldrich, St. Louis, MO, USA), 0.02% bovine serum albumin (SERVA Electrophoresis GmbH, Heidelberg, Germany), and penicillin/streptomycin (GE Healthcare, South Logan, UT, USA). Cells were grown at 37 °C in humidified air containing 5% $CO_2$. Cell medium was exchanged every 48 h.

### 2.2. Plasmids

Plasmid mVenus C1, which is 4731 bp in size (Addgene plasmid #27794, a gift from Steven Vogel) or 3486 bp long pmaxGFP™ vector (supplied in Nucleofector™ Kits) were used as a plasmid-containing reporter fluorescent protein for estimation of transfection condition and efficiency at 100% of total DNA. Empty gRNA plasmid is 3914 bp in size (Addgene plasmid #41824, gRNA_Cloning Vector, a gift from George Church) (75% of total DNA) and dCas9-Dnmt3a3L-KRAB plasmid (11,548 bp in size) (constructed by Dr. Tomasz Jurkowski) (20% of total DNA), combined with 5% of reporter plasmid were used for transfection.

### 2.3. Polyethylenimine (PEI)-Based Cell Transfection

The working solution of PEI was made in water at the concentration of 1 mg/mL, filter-sterilized through a 0.22 μm membrane, aliquoted, and stored at −80 °C until needed. Cells ($1.5 \times 10^6$) were seeded in a six-well plate in a complete medium and incubated until they grew to a confluence of approximately 60%. After washing with phosphate buffer solution (PBS), cells were transfected by adding a mix of DNA and MAX PEI (Polysciences Inc., Warrington, PA, USA) to the cells in PEI to DNA mass ratio of 3:1. Before addition to the cells, plasmid DNA (2 μg) and PEI (6 μg) were each diluted with equal volumes of 15 mM glucose DMEM without FBS, mixed, and incubated for 20 min at room temperature (RT). After 16 h of incubation at 37 °C, cells were washed in PBS and immersed in a complete cell medium. Subsequently, the medium was exchanged every two days during the experiment.

Protocol for Chemical Transfection

- Preparatory work

  ○ Two days before transfection, plate $1.5 \times 10^6$ into six-well plates in DMEM.
  ○ Warm PBS, complete and incomplete DMEM (without FBS) to 37 °C.

- Mix plasmid DNA in incomplete DMEM media at a ratio of 2 μg DNA in 100 μL media for transfection in a six-well plate. Incubate 5 min at RT.
- Add 6 μL of PEI (1 mg/mL of working solution) per 100 μL of incomplete DMEM. Mix them by tapping and incubate for 5 min at RT.

- Add DNA to PEI drop-wise with constant tapping of the cuvette. Incubate for 20 min at RT.
- Add the final mix of DNA:PEI drop-wise to the cells with 1.8 mL complete DMEM.
- Gently rock the plate with the cells for uniform distribution of mix for transfection.
- Incubate at 37 °C for 16 h.
- Rinse the cells with PBS, and propagate cells until analysis.

### 2.4. Nucleofection

For nucleofection, αTC1-6 cells were grown to a confluence of 70–80%. Cells were washed in PBS and detached by cell dissociation, enzyme-free, PBS buffer (GIBCO, by Life Technologies). The optimization protocol was followed according to the manufacturer's recommendations (Amaxa™ 4D-Nucleofector™ Optimization Protocol for Cell Lines). The required cell number ($3.4 \times 10^6$ cells per 4D-Nucleofector™ X Solution) was centrifuged at $90 \times g$ for 10 min at RT. Following the centrifugation step, cell pellets were resuspended in 340 µL of SE, SF, or SG 4D-Nucleofector™ X Solution with a supplement at RT ($2 \times 10^5$ cells in 20 µL 4D-Nucleofector™ X Solution with a supplement for one well in 16-well Nucleocuvette™ Strips, if not specified otherwise). The supplement was added in a *v:v* ratio of 1:4.5. Afterward, the cell suspension was mixed with 0.4 µg pmaxGFP™ (Lonza, 1 µg/µL in 10 mM Tris pH 8.0) plasmid DNA (if not indicated otherwise), transferred into the 16-well Nucleocuvette™ Strips and electroporated by using the 4D-Nucleofector™ X Unit (Lonza). Optimization of nucleofection included 15 different nucleofector programs and no program control. After nucleofection, cells were preequilibrated with RPMI for 10 min at 37 °C and then transferred into a prepared culture dish with a prewarmed DMEM medium. The protocol for optimization of nucleofection is available at https://bioscience.lonza.com/lonza_bs/US/en/download/content/asset/21451 (accessed on 4 February 2022).

Optimized Protocol for Nucleofection for the αTC1-6 Cell Line

- After reaching 70% of cell confluency, aspirate the medium from the flask. Wash the cells once with PBS and harvest them by using cell dissociation buffer.
- Determine cell density.
- Centrifuge $5 \times 10^6$ cells per one 100 µL single Nucleocuvette™ at $90 \times g$ for 10 min at RT and completely remove the supernatant.
- Add 100 µL 4D-Nucleofector™ SF Solution with supplement and 7.5-µg plasmids on dry cells pellet. Mix gently with pipetting.
- Transfer cells into the 100-µL single Nucleocuvette™.
- Place Nucleocuvette™ into the retainer of the 4D-Nucleofector™ X Unit and start the CM-156 nucleofection program.
- After run completion, add 400 µL of RPMI medium to each Nucleocuvette™ and incubate them for 10 min at 37 °C.
- Gently resuspend cells by pipetting two to three times and transfer cells in preincubated six-well plates with complete DMEM.
- Propagate cells until analysis.

### 2.5. Fluorescent Microscopy

After PEI-based cell transfection and nucleofection, cells were propagated for mostly 5–8 days and checked for GFP expression at several time points by fluorescent microscopy. Images were taken with an Axiocam digital camera attached to the Axio Observer Z1 microscope (Carl Zeiss Microscopy GmbH, Jena, Germany) by using appropriate filters. Quantification of fluorescence was done by open-source software Image J (version 1.52, Wayne Rasband, National Institutes of Health, Bethesda, MD, USA). Integrated density sums up all of the pixels within a region and gives a total value of the fluorescence signal.

## 2.6. Flow Cytometry and Cell Sorting

The expression of GFP in transfected cells at three different time points (24 h, 5 and 7 days after PEI-based cell transfection and nucleofection) was tested by flow cytometry. Cells were analyzed on Partec CyFlow Space flow cytometer (Partec, Munster, Germany) by using FlowMax software or FACS Aria III flow cytometer and cell sorter (BD Biosciences, San Diego, CA, USA) using FACS Diva software. On the fifth and seventh day after transfection, cells were subjected to fluorescence-activated cell sorting for separating and collecting GFP positive cells by using FACS Aria III flow cytometer and cell sorter. Cell samples were resuspended in Hanks' buffered saline solution (HBSS) buffer without calcium and magnesium ions enriched with 2% FBS and 2 mM EDTA.

## 2.7. RNA Isolation and Real-Time Quantitative PCR (RT-qPCR)

The ZR-Duet™ DNA/RNA MiniPrep Kit (Zymo Research, Irvine, CA, USA) was used to isolate total RNA from sorted αTC1-6 cells on the fifth and seventh day after transfection. For complementary DNA synthesis, total RNA was treated with DNAse I and reverse-transcribed with RevertAid First Strand cDNA Synthesis Kit (Thermo Fisher Scientific, Waltham, MA, USA) by using mixed oligo(dT) and random primers (1:1). The levels of mRNA were quantified by RT-qPCR by using Maxima SYBR Green/ROX qPCR Master Mix (2×) (Thermo Fischer Scientific, Waltham, MA, USA) and the QuantStudio 3 Real-Time PCR system (Applied Biosystems, Carlsbad, CA, USA). The thermal cycles included an initial denaturation step (95 °C /10 min) and 40 cycles of two-step PCR at 95 °C /15 s and 60 °C /60 s. The relative expression levels of the Cas9 target gene were calculated by using the comparative $2^{-\Delta\Delta Ct}$ method after normalization by using receptor accessory protein 5 (REEP 5) as an endogenous control. Primers used for fragment amplification were:

Cas9: Fw 5′–TCAGGCGGCAAGAGGATTTC-3′, Rev 5′-AGTCATCCACGCGAATCTGG-3′, REEP 5: Fw 5′-TCATCGGACTGGTGGCTTTG-3′, Rev 5′-GTTGGGACTCTCGATGGCTT-3′.

## 2.8. Immunocytochemistry

After nucleofection, cells were propagated on sterile glass coverslips in 24-well sterile culture plates. On the fifth and seventh day post-transfection (dpt), cells were fixed in 4% paraformaldehyde (Science Services GmbH, Munich, Germany) in PBS for 10 min at RT, permeabilized in the 0.3% Triton X-100 in PBS for 10 min at RT, and blocked in 3% bovine serum albumin in PBS for 1 h at RT. The coverslips were incubated overnight at 4 °C with an anti-glucagon antibody (C-18, Santa Cruz Biotechnology, Santa Cruz, CA, USA) diluted at 1:50 in 0.2% Tween-20 in PBS. Fluorescently labeled secondary antibody (sc-3855, donkey anti-goat, tetramethyl rhodamine iso-thiocyanate (TRITC), Santa Cruz Biotechnology, Santa Cruz, CA, USA) was diluted 1:100 in 0.2% Tween-20 in PBS and incubated 2 h at RT with coverslips. All wash steps were completed in 0.2% Tween-20 in PBS. DNA was visualized by adding 4,6-diamidino-2-phenylindole (DAPI) (Roche Diagnostics, Mannheim, Germany) (0.1 μg/mL) for 5 min at RT. The coverslips were glued to the glass slides with mowiol (Calbiochem, San Diego, CA, USA). An Axio Observer Z1 microscope (Carl Zeiss Microscopy GmbH, Jena, Germany) with the Axiocam digital camera was used for taking images, by an appropriate filter.

## 2.9. Statistical Analysis

All the data were analyzed with GraphPad Prism 5 software for Windows (GraphPad Software, La Jolla, CA, USA, www.graphpad.com (accessed on 7 July 2022)). Experiments were done in three biological replicates and presented as mean values ± SDs. A two-tailed unpaired Student's *t*-test was used to compare the mean values of the variables between the two groups. One-way ANOVA followed by Tukey's HSD multiple comparison post hoc test was used for comparison of the mean values of more than two analyzed groups. A *p*-value less than 0.05 was considered statistically significant.

## 3. Results and Discussion

### 3.1. PEI-Based Transfection of αTC1-6 Cells

The rapidly developed genome engineering technologies, such as CRISPR/Cas9, offer many possibilities for application in research. For efficient (epi)genome editing, the imperative factor is efficient delivery in the host cells. Transfection is a widely used method for the delivery of foreign nucleic acid or protein in eukaryotic cells which allows for studying the regulation of gene expression and the function of gene products by enhancing or inhibiting its expression [9]. As each mammalian cell type has a specific set of requirements for efficient transfection, there is a tremendous need to optimize the protocol for cell transfection. Depending on the effect that the epigenetic manipulation in the cell expects to ensure, there is a need for a time course plan of the experiment. In this research, the mouse pancreatic cell line αTC1-6 was used as a model system for epigenetic manipulation. This cell line is difficult to propagate, cells are sensitive to rough manipulation and oxygen level. Our main goal was to find a satisfying transfection condition that will provide a sufficient mass of transfected cells at a few points during the time course.

One of the widely used methods for introducing foreign DNA into the host cell is transfection by using PEI [22]. This type of chemical transfection is considered the gold standard because PEI is powerful, easy to use, quick, cost-effective, and shows low cytotoxicity. This cationic polymer condenses DNA into stabile, positively charged particles that bind to anionic cell surfaces where they have been endocytosed and released in the cell cytoplasm [23]. For the first step, small reporter plasmid mVenus C1 was used, and the most common PEI:DNA ratio (3:1) to examine transfection efficiency (Figure 1). Cells were propagated until a fifth dpt, and no negative effects on cell viability and proliferation were observed. Transfected cells behaved and looked equally good as untransfected cells. Moreover, they showed a statistically higher number of cells, indicating a better proliferation rate compared to the control untransfected cells (Figure 1A). The GFP expression level was measured as a marker for transfection efficiency [24]. Flow cytometry showed ~30% GFP positive cells which was encouraging regarding untouched cell viability (Figure 1B,C). Moreover, cells were visualized by fluorescent microscopy on the fifth dpt to check the retention of plasmid DNA through cell division (Figure 1D). This level of transfection efficiency was satisfying, so for the next step, cells were transfected with an EpiCRISPR plasmid mix.

In the next step, cells were co-transfected with a mix of plasmids: 5% mVenus C1 (as a reporter gene), 20% dCas9-Dnmt3a3L-KRAB plasmid (as effector domain), 75% empty gRNA plasmid (in this study, the plasmid lacked the information for targeted genome region but in the future, it will carry the designed sequence for gRNA) with the same PEI:DNA ratio as previously described. The same transfection reagent, PEI, was used by for transfection of the FACE plasmid in different mammalian cell lines with high transfection rate and cell viability upon transfection [25]. On the fifth dpt, the flow cytometer and cell sorter were used for measuring and collecting transfected GFP positive cells. As proof of EpiCRISPR system delivery, the relative level of Cas9 mRNA was measured in the sorted cell fraction (Figure 2C). The detected level of GFP positive cells by fluorescent microscopy and FACS (Figure 2A,B) was very low (around 3.8%) relative to previous results (Figure 1). The potential reason for weak transfection efficiency could be the large size of the EpiCRISPR plasmid which is 11,548 bp. The same assumption was drawn by Lesueur, et al. in 2016, who showed that transfection efficiency is inversely correlated to the plasmid size 24 h after plasmid electrotransfer in adipose tissue-derived mesenchymal stem cells [21].

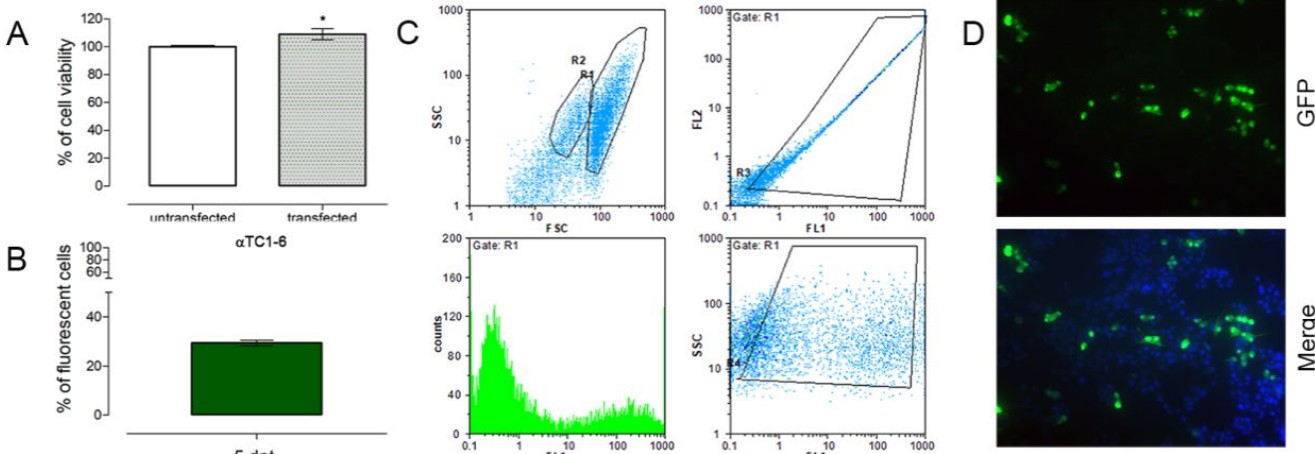

**Figure 1.** -Transfection of αTC1-6 cells performed using PEI. Cells were transfected with mVenus plasmid vector in 1:3 = DNA:PEI ratio. (**A**) Cell viability and (**B**) percent of fluorescent cells was observed fifth day post-transfection (dpt) by flow cytometry. (**C**) Flow cytometric acquisition of ~30% of GFP positive cells. (**D**) Representative images of efficient αTC1-6 transfection on the 5th dpt. Cell nuclei were stained with DAPI. The results are expressed as means ± SDs. Significance between samples was determined by using an unpaired Students t-test, * $p \leq 0.05$.

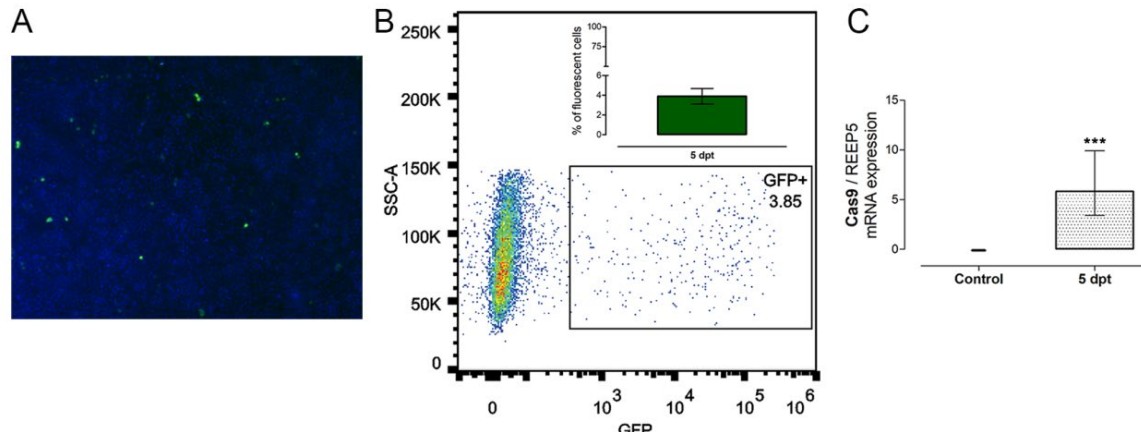

**Figure 2.** PEI transfection of αTC1-6 cells with the EpiCRISPR mix of plasmids. Cells were transfected with a plasmids mix: 5% Venus C1, 20% dCas9-Dnmt3a3L-KRAB, and 75% empty gRNA plasmid. (**A**) Fluorescent microscopy showed very few fluorescent cells on the fifth dpt. (**B**) Flow cytometry showed that there was ~3.8% of transfected GFP positive cells. (**C**) Relative expression level of Cas9 mRNA in GFP positive sorted cells on the fifth dpt was measured by RT-qPCR. The results are expressed as means ± SDs. Significance between samples was determined by using an unpaired Students *t*-test, *** $p < 0.0001$.

### 3.2. Nucleofection of αTC1-6 Cells

Furthermore, we used a nucleofection as an attractive alternative transfection method that could offer high efficiency of transfection level. Nucleofection is a type of electroporation that represents a widely used physical transfection method. As a result of an applied electrical pulse, transient pores are formed in the cell and nuclear membrane allowing for a free pass of target molecules [26]. In this way, nucleic acids enter directly into the nucleus, which increases the efficiency of transfection. This method provides faster gene expression because there is no need for cell division, which is very important for non-dividing primary cells.

In our experiments, Nucleofector® Technology was the method of choice due to its higher efficiency for large DNA-fragment transfection and increasing cell survival rate in comparison with the electroporation [27,28]. Also, nucleofection allows multiple samples to be transfected at the same time, making it a higher throughput method than electroporation. There was no ready-to-use optimized protocol or published data for αTC1-6 cells using this transfection tool; therefore we start to analyze the best conditions for effective transfer of plasmid molecules in cells which involves a balance between the highest efficiencies of gene transfer and the minimal amount of cell death [19] (Figure 3). The specific combination of predefined pulses and three different solutions were used with the unknown supplement composition in combination with the pmaxGFP™ control vector, which encodes the green fluorescent protein from *Copepod Pontellina* sp., all provided from Cell Line Optimization 4D-Nucleofector™ X Kit. The SF buffer stood out as the best among three different solutions (SE, SF, and SG) in combination with 15 different predefined Nucleofector™ programs. For this experiment, we used 0.4 μg of pmaxGFP™ for $2 \times 10^5$ αTC1-6 cells suspended in 20 μL SF, SG, or SE 4D-Nucleofector X Solution with a supplement for one well in 16-well Nucleocuvette™ Strips as was recommended. After nucleofection, cells were immediately seeded in four 96-wells ($0.5 \times 10^5$ per well). Cells that were in SF buffer show the best percentage of viability, divided into two groups: one with high survival level and the other with a lower survival level. Compared to the other two buffers, the SF buffer had the least impact on cell membrane disruption, enabling most of the cells to survive and successfully recover after applied electrical pulses (Figure 3A). Also, these cells showed the best percentage of GFP positive cells (~50% GFP positive cells) (Figure 3B). On the other hand, the SE buffer showed the worst results concerning αTC1-6 cells' viability, resulting in a low number of transfected cells.

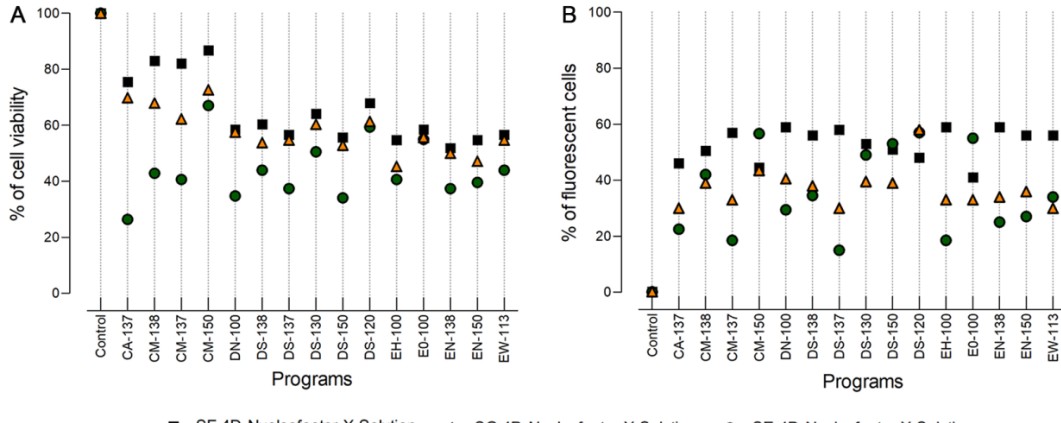

**Figure 3.** Comparison of three 4D-Nucleofector X Solutions and predefined programs for αTC1-6 cells nucleofection. (**A**) Percentage of cell viability 24 h after nucleofection was observed by flow cytometry. Cells with no program were taken as a non-transfected control cells and were 100% viable. Percentage of nucleofected cells is expressed relative to control cells. (**B**) Percentage of GFP positive cells 24 h after nucleofection was observed by flow cytometry.

Based on this result, in the next optimization experiment we used exclusively SF 4D-Nucleofector X Solutions (Figure 4). By using available recommendations, we increased the number of cells per transfection and included a recovery step with incubation in a low calcium medium (RPMI) for 10 min at 37 °C after nucleoporation.

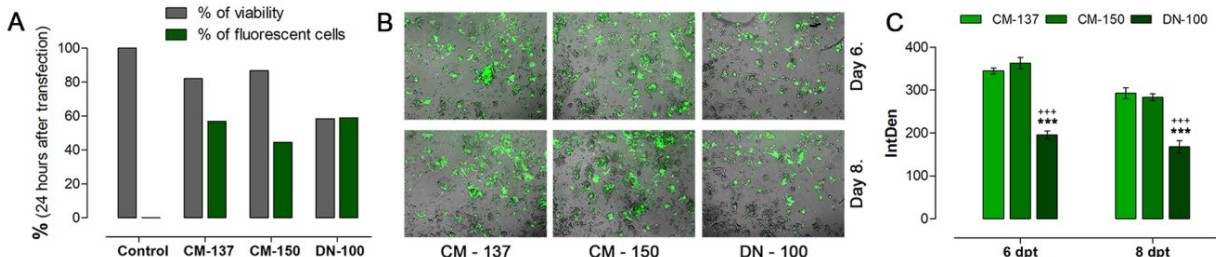

**Figure 4.** Comparison of the most potent combinations of SF 4D-Nucleofector X Solutions and nucleofection programs for αTC1-6 cells. (**A**) The three most potent programs at 24 h after transfection are selected for monitoring on the sixth and eighth days after nucleofection. Control represents non-transfected cells. (**B**) Post-nucleofection time course of GFP expression level is monitored by using fluorescent microscopy. Representative photos of selected programs for nucleofection are shown. (**C**) The results of quantification of the fluorescent signal are expressed as means ± SDs. Significance between groups was determined by using one-way ANOVA followed by Tukey's post hoc test. *** $p < 0.0001$ compared to CM-137, +++ $p < 0.0001$ compared to CM-150.

Programs CM-137 and CM-150 were chosen between four programs (CA-137, CM-138, CM-137, CM-150) with the ability to provide the highest percentage of cell viability (around 80%), where CM-137 provided the highest and CM-150 the lowest transfection efficiency. Among the other 11 programs that induced lower viability after transfection (around 58%), DN-100 was chosen as a program providing the highest efficiency and very good survival rate (Figure 3). Cells exposed to CM-137 and CM-150 programs showed a similar rate of survival 24 h after nucleoporation whereas program DN-100 caused larger cell damage affecting the cell viability by 41.51% (Figure 4A). On the other hand, CM-137 and DN-100 programs gave a similar rate of transfection efficiency (CM-137–57%, DN-100–59%) whereas the CM-150 program induced a slightly lower efficiency rate (44.5%). GFP expression levels were examined on the sixth and eighth days after transfection for three selected pulses. The maintenance of GFP expression was noticed until cells were tracked (a few GFP positive cells on 14dpt, data not shown). The cells exposed to the CM-150 program showed the best viability 24 h after nucleoporation and had the most successful recovery after stress caused by an electrical pulse (Figure 4B,C). This was indicated with initial low mortality after electrical pulse and level of cell divisions after nucleofection, followed by maintaining a high level of GFP fluorescence. Both CM-137 and CM-150 programs resulted in significantly more cells expressing GFP than the DN-100 program (Figure 4C), which highlights the importance of low cytotoxicity of the transfection method for obtaining sufficient pull of transfected cells for further analysis. Further optimization experiments were directed toward the increasing pull of transfected cells. There were two directions: one involved increase in the initial cell number for transfection and the other an increase of the amount of plasmid DNA (Figure 5).

In order to increase the number of transfected cells, the effect of cell seeding density on cell viability was examined (Figure 5). As shown in Figure 3, $2 \times 10^5$ cells were taken as 100% viable. We observed that doubling the cells' number had no adverse effect on cell viability 24 h after seeding. An additional significant increase of cell numbers harmed cell viability because high initial cell densities might have a negative effect on cell metabolism and nutrient availability (Figure 5A). Furthermore, nucleofection optimization included a comparison of CM-137 and CM-150 programs in order to choose the most appropriate transfection conditions (Figure 5B,C). For each program, we used $3.5 \times 10^5$ αTC1-6 cells and 0.5 μg pmaxGFP™ plasmid. Cells were suspended in 20 μL SF 4D-Nucleofector X Solutions, including incubation in RPMI medium for 10 min after nucleoporation at 37 °C as a recovery step. This recovery step was included to prevent calcium influx from media to the cell. Transient pores generated after nucleofection remain open for about 15 min, enabling calcium ions from the DMEM medium containing high levels of calcium ions to enter the cell. This calcium influx could affect cell viability through the initiation of

many signal transduction pathways. RPMI medium with a low level of calcium ions was used as a "recovery medium," supporting the cell's survival. After increasing the cell number for nucleofection by 75%, cell viability was measured 24 h after transfection. The viability of cells exposed to the program CM-137 were 17% lower if compared with the previous experiment (Figure 3A). The viability of the cells subjected to program CM-150 stood nearly the same as previously (89.9%) and showed a statistically significant higher rate of survival compared to the CM-137 program (Figure 5B). The additional recovery step after the electrical pulse had a beneficial effect on the increasing pull of transfected cells by increasing the number of GFP-positive cells by 8.6% for cells exposed to program CM-150, whereas there were no significant changes for cells exposed to program CM-137 (Figure 5C) compared to the previous experiment (Figure 3B).

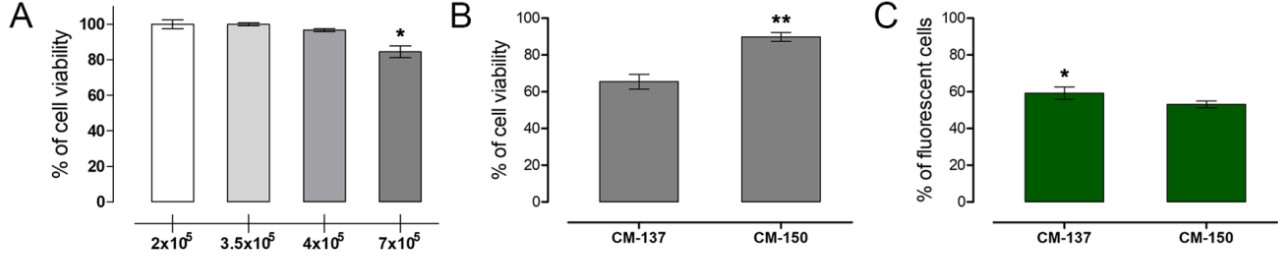

**Figure 5.** Increased cell number enhances nucleofection outcomes. (**A**) Cell viability was observed by flow cytometry and cell viability at cell density of $2 \times 10^5$ has been taken as a 100%. The results are expressed as means $\pm$ SDs. Significance between groups was determined by using one-way ANOVA followed by Tukey's post hoc test, * $p \leq 0.05$ compared to all three groups of cells. (**B**) Cell viability and (**C**) the percent of GFP-positive cells are measured by flow cytometry. For monitoring nucleofection efficiency, $3.5 \times 10^5$ cells were transfected with 0.5 μg of plasmid DNA pmaxGFP™ by using the two most potent predefined programs for nucleofection (CM-137 and CM-150). The results are expressed as means $\pm$ SDs. Significance between samples was determined by using an unpaired Students $t$-test, * $p \leq 0.05$, ** $p \leq 0.01$.

Our results revealed that nucleofection is a method of choice for αTC1-6 cells transfection, because we obtained a better transfection rate and higher cell viability for αTC1-6 cells in compare to the DNA:PEI transfection method (Figures 1 and 2).

The next step in nucleofection experiments included the use of the EpiCRISPR mix of plasmids: 5% pmaxGFP™ (as a reporter gene), 20% dCas9-Dnmt3a3L-KRAB plasmid (as effector domain), and 75% empty gRNA plasmid in two different final amounts (0.5 μg and 1 μg) for $3.5 \times 10^5$ cells per pulse (Figure 6).

Twenty-four hours after the nucleofection, flow cytometry analysis showed that doubling the amount of EpiCRISPR mix of plasmid DNA did not affect the viability of the cells exposed to both programs CM-137 and CM-150 (Figure 6A). The cell survival rate was the same as in the previous experiment (Figure 4A) confirming the reproducibility of this tool for cell transfection. Despite the high degree of cell viability, transfection efficiency was affected by the size of plasmids (Figure 6B) as was noticed with DNA:PEI transfection (Figures 1 and 2). The pmaxGFP™ vector belongs to the group of the small reporter plasmids and is 3486 bp in size, whereas dCas9-Dnmt3a3L-KRAB is almost $3\times$ larger (11,548 bp in size). During the transfection process, larger plasmids can become tangled into open membrane pores, aggravating proper plasmid uptake and membrane reclosure and contributing to cell death [29]. As shown in Figure 6B, transfection efficiency was satisfying but lower than in previous experiments for 27.3% if CM-137 was applied and 33.6% if CM-150 was applied, because of co-transfection of αTC1-6 cells with a mix of larger plasmids (5% pmaxGFP™, 20% dCas9-Dnmt3a3L-KRAB plasmid and 75% empty gRNA plasmid). As stated by others and similar to what we observed, through co-traveling across the cell and nuclear membrane, small plasmids facilitate the entrance of larger plasmids like

dCas9-Dnmt3a3L-KRAB plasmid leading to improved transfection efficiency [30]. The drop in transfection efficiency is much slighter after nucleofection compared to the DNA:PEI transfection method (Figure 2A) because the size of the plasmid has a lesser impact on transfection efficiency when a larger plasmid is used in combination with a small one (as it is in our study). Moreover, the increase in the total amount of plasmid DNA leads to twice the transfection efficiency (Figure 6B). Program CM-150 was selected for further optimization experiments because earlier it showed better cell viability after nucleofection with pmaxGFP™ DNA plasmid relative to CM-137 (Figure 5B,C).

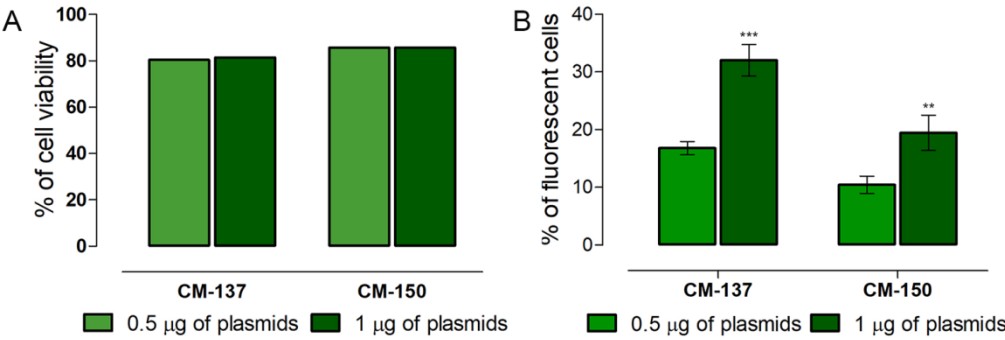

**Figure 6.** Optimization of the selected programs and DNA amount for nucleofection with EpiCRISPR plasmids. (**A**) Cell viability and fluorescence were observed 24 h after seeding by flow cytometry. (**B**) Transfection efficiency was estimated as percentage of fluorescent cells. Nucleofection was performed with two different plasmid concentrations. The results are expressed as means ± SDs. Significance between samples was determined by using an unpaired Students *t*-test. ** $p \leq 0.01$, *** $p < 0.0001$.

Although we observed improved DNA transfection efficiency with the nuclear transfer of plasmids by nucleofection in comparison with DNA:PEI transfection, fine-tuning of the selected pulse was done to further increase transfection efficiency (Figure 7). For this experiment, we used 2 µg (mix of plasmids: 5% pmaxGFP™, 20% dCas9-Dnmt3a3L-KRAB plasmid, 75% empty gRNA plasmid) of plasmid DNA and transfected $5 \times 10^5$ αTC1-6 cells suspended in 20 µL SF 4D-Nucleofector X Solutions.

The program CM-156, which is designed for higher efficiency, was selected from the fine-tuning matrix (Figure 7C) and compared with CM-150 in an attempt to increase the number of GFP positive cells. On the second day after transfection, cells were observed by fluorescent microscopy (Figure 7A). On the fifth day after transfection was chosen as the next time point, because it is an optimal time frame for the EpiCRISPR system to make necessary changes in the epigenome and for those changes to become measurable [31].

An increase in the percentage of transfected cells (Figure 7B), if compared to the previous experiment after nucleofection using the CM-150 program (Figure 6B), could be a consequence of the doubling of plasmid DNA (from 1 to 2 µg) as well as increasing the cell number per transfection (from $3.5 \times 10^5$ to $5 \times 10^5$). Presented results indicated that the CM-156 program provided an improved effect on transfection efficiency as can be seen by the amount of GFP positive cells with a limited effect on cell growth.

For transient transfection, introduced genetic material is not integrated into the genome and is expressed only for a restricted period, because it can be lost by environmental factors and cell division [19]. The optimal time point for measuring the effect of introduced DNA into the nucleus of the cells is between the first or fourth dpt [9]. However, when transfection serves for delivery of the epigenome-editing machinery into the cells, effects of introduced epigenetic changes like DNA methylation, could require more time to make the physiological effects. The selected program CM-156 was tried in large-scale cuvettes, and the results were scaled and reproducible (Figure 8). For this experiment, cells were nucleofected in 100-µL Nucleocuvette™ vessels with a 7.5 µg of plasmids mix (5% pmaxGFP™, 20% dCas9-Dnmt3a3L-KRAB plasmid, 75% empty gRNA plasmid). We

used $5 \times 10^6$ cells per Nucleocuvette™. The nucleofection shows much better results in transfection efficiency than chemical DNA:PEI transfection for the analyzed cell line.

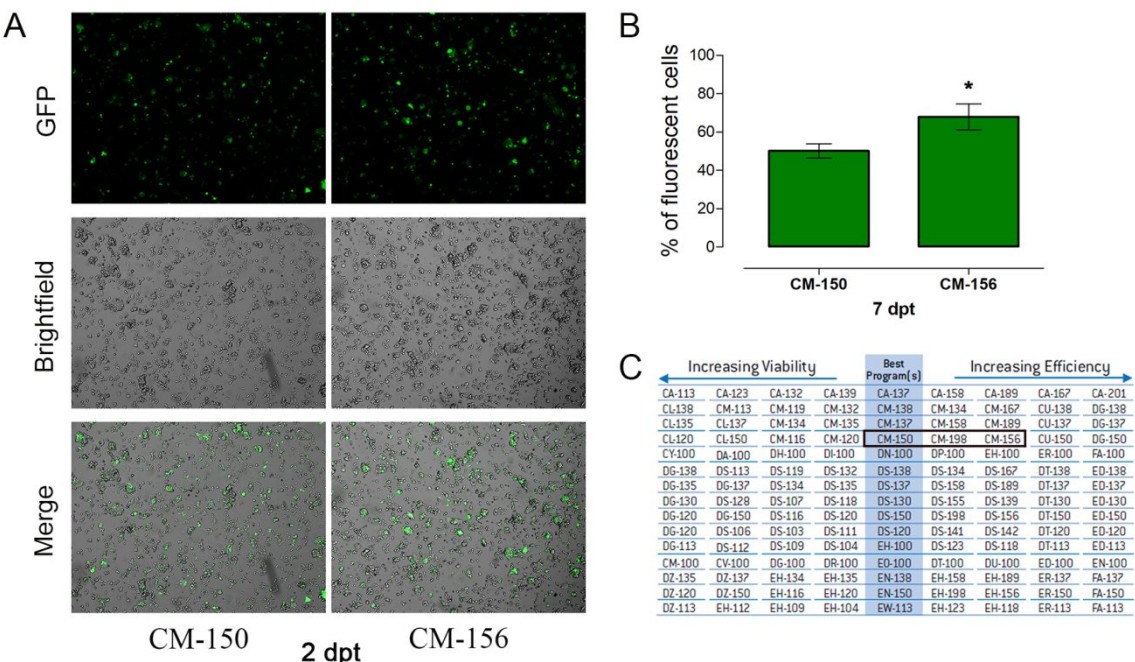

**Figure 7.** Fine-tuning of the selected pulse. Transfection efficiency was observed 48 h after seeding $5 \times 10^5$ by fluorescent microscopy (**A**) and by flow cytometry on the fifth post-transfection day (**B**). (**C**) In an attempt to increase the number of GFP positive cells, the CM-150 and CM-156 program have been moved in a direction toward increasing efficiency. Fine-tuning matrix available at https://bioscience.lonza.com/download/content/asset/31152 (accessed on 4 February 2022). The results are expressed as means $\pm$ SDs. The significance between samples was determined by using an unpaired Students *t*-test, * $p \leq 0.05$.

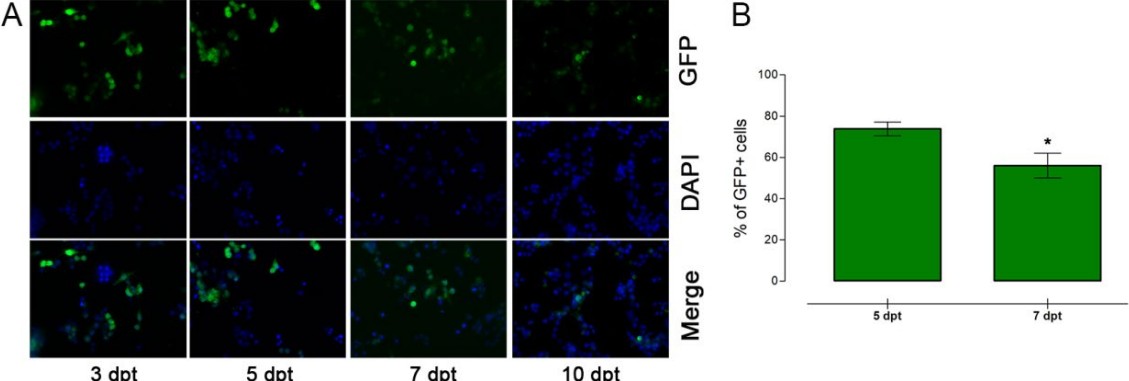

**Figure 8.** Monitoring of GFP expression level in cells nucleofected with selected CM-156 program. (**A**) Post-nucleofection time course of GFP expression level was monitored via fluorescent microscopy. Cell nuclei are stained with DAPI. (**B**) Flow cytometry analysis showed a high level of GFP expression on the fifth day after nucleofection, followed by a reduction of GFP expression on the tenth post-nucleofection day. The results are expressed as means $\pm$ SDs. The significance between samples was determined by using an unpaired Student *t*-test,* $p \leq 0.05$.

Fluorescent microscope analysis showed the variable expression levels of the GFP at several time points indicating time-limited transient transfection of αTC1-6. Ten days after nucleofection GFP is still detectable in the cells, but at a very low level compared to the other dpts (Figure 8A). Cells were analyzed and sorted by FACS on the fifth and seventh

days after transfection. The detected level of GFP-positive cells was around 71.12% for the fifth and 59.5% for the seventh dpt (Figure 8B). The drop in the percentage of positive cells between the fifth and seventh day is a consequence of losing the plasmids during ongoing cell division and their degradation.

The main function of alpha pancreatic cells is a secretion of glucagon which participates in maintaining glucose homeostasis [32]. After transfection, there were no differences in the presence of glucagon in the group of transfected cells relative to the control (untransfected) $\alpha$TC1-6 cells (Figure 9A). This indicates that the presence of the EpiCRISPR system in the transfected cells without gRNA for targeting the region of interest did not affect their main function. The relative level of Cas9 mRNA confirmed successful entrance and presence of EpiCRISPR system in transfected and collected GFP-positive $\alpha$TC1-6 cells at the fifth and seventh dpt (Figure 9B). The change in expression of the Cas9 mRNA level between two analyzed days is correlated with the change in expression of green fluorescent protein in transfected cells, demonstrating transient transfection of $\alpha$TC1-6 cells. The same was observed by Subramanian and Srienc, who noticed the specific growth rate was in correlation with the rates of the decrease of fluorescence in transiently transfected CHO cells [33].

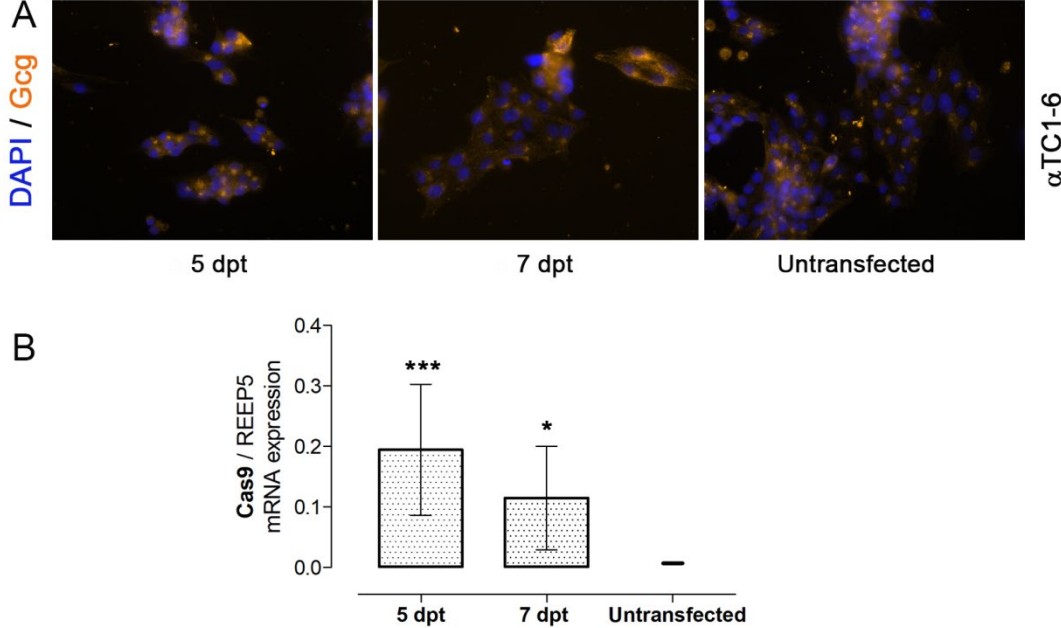

**Figure 9.** The functionality of $\alpha$TC1-6 was not disturbed after nucleofection. (**A**) Cells were immunostained with the anti-glucagon antibody on the fifth and seventh days after transfection. Cell nuclei were stained with DAPI. (**B**) Relative expression level of Cas9 mRNA in control untransfected cells and in cells on the fifth and seventh days after transfection measured by RT-qPCR. The results are expressed as means $\pm$ SDs. Significance between samples was determined by using one-way ANOVA followed by Tukey's post hoc test. * $p \leq 0.05$, *** $p \leq 0.0001$ compared to untransfected cells.

Different transfection methods and reagents have to be defined for cells that are hard to transfect. The CRISPR/Cas system could be delivered to target cells by using numerous methods for cell transfection [34]. Pini et al. screened several non-viral methods (Lipofectamine2000 (Thermo Fisher Scientific, Waltham, MA, USA), TurboFect (Thermo Fisher Scientific, Waltham, MA, USA), GeneJuice (Sigma-Aldrich, St. Louis, MO, USA) and electroporation (Thermo Fisher Scientific, Waltham, MA, USA)) to deliver CRISPR/Cas9 in primary myoblasts, showing that electroporation has at least 2.5$\times$ higher transfection efficiency than other chemically based methods [35]. Electroporation performed with a Nucleofector™ II/2b device was selected for cotransfecting plasmids encoding all elements for CRISPR/Cas system together with piggyBac transposon system for delivery and ex-

pression CAR gene [36]. For malignant B cells, which are extremely difficult to transfect, Nucleofector™ technology-based electroporation system was used to make ROR1 knockout cells by using the CRISPR/Cas9 system [37]. Recent approaches used electroporation for successful delivery of Cas9 ribonucleoprotein (RNP) complexes consisting of the Cas9 protein and a fluorescently labeled crRNA/tracrRNA duplex targeting genes [38–40]. As shown in the paper of Savell et al. (2019) the CRISPR-based construct (sgRNA:dCas9-VPR) was delivered to HEK293T cells by using FuGene HD (Promega) but delicate, C6 glioma cells were transfected by using a Nucleofector 2b device (Lonza) [41]. The αTC1-6 cells are adherent and grow as loosely attached clusters with some single cells in suspension. PEI-mediated cell transfection is not a suitable method for the cells growing in clusters because there is a part of the exposed cells to DNA/PEI complexes, whereas the rest of the cells are unable to be in contact with the DNA/PEI complexes due to their clustering tendency. Therefore, in our study, nucleofection was chosen as the transfection method for αTC1-6 cells, because we observed that these cells are harsh to maintain in culture, are growing in clusters, and are very sensitive to the lack of $CO_2$; therefore we foresee problems with transfection, especially because our DNA construct is rather large (~11 kb). Regarding the transfection efficiency, an approximately 40% nucleofection efficiency was observed earlier for the αTC1-6 cell line transfected with the pCMV-3FLAG-3a plasmid carrying the cloned gene upstream (5′) of the triple-FLAG repeat moiety. In this experiment, nucleofection was performed on an Amaxa 2b device [42]. After optimization, we succeeded to achieve approximately 70% nucleofection with the Amaxa 4D X unit. Finally, one of the reasons for better efficiency of nucleoporation could relay in the use of cell suspension whereby all cells have an equal chance to be transfected. All of this together emphasizes nucleofection as a major technique for the efficient delivery of EpiCRISPR in vitro in αTC1-6 cells.

**Author Contributions:** Conceptualization, M.V. and J.A.J.; methodology, M.Đ., M.J.T., and V.P.; validation, M.Đ. and J.R.; formal analysis, M.Đ., A.T., and A.U.; investigation, M.Đ. and M.M.; writing—original draft preparation, M.Đ.; writing—review and editing, M.V. and J.A.J.; visualization, M.Đ., N.G., and S.D.; supervision, J.A.J.; funding acquisition, M.V. and I.M. All authors have read and agreed to the published version of the manuscript.

**Funding:** This work was supported by the European Foundation for the Study of Diabetes (EFSD), European Diabetes Research Programme in Cellular Plasticity Underlying the Pathophysiology of Type 2 Diabetes, research grant from AstraZeneca (M.V.) and by the Ministry of Education, Science and Technological Development of the Republic of Serbia (Grant No. 451-03-68/2022-14/200007 and 451-03-68/2022-14/200110). Furthermore, this article is partially based upon work from COST Actions TD0509, CM1406 and CA16119, supported by COST (European Cooperation in Science and Technology).

**Institutional Review Board Statement:** Not applicable.

**Informed Consent Statement:** Not applicable.

**Data Availability Statement:** Not applicable.

**Acknowledgments:** The authors are very grateful to: Tomasz P. Jurkowski for his professional help and friendship and for providing us dCas9-Dnmt3a3L-KRAB plasmid; Steven Vogel for providing mVenus C1 plasmid and to George Church for empty gRNA plasmid. In addition, thanks goes to Djordje Miljković for his help in flow cytometric analysis on Partec.

**Conflicts of Interest:** The authors declare no conflict of interest.

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
