# Peer review of "Nucleofection as an Efficient Method for Alpha TC1-6 Cell Line Transfection"

_applsci, doi:10.3390/app12157938_

Round 1

Reviewer 1 Report

I would recommend removing PEI results and comparison and stand alone optimisation of nucleofection protocol. To conclude whether physical or chemical methods is better for a cell line, more than two methods are required. Even within the electroporation, Neon electroporation may perform better than nucleofection.  It all depends on the cell types and cargo.

There is error in figure 1B legend.

Reviewer 2 Report

The title, the summary and keywords adequate and well suited.

Introduction – line 63 (The chemical – new paragraph); line 74 (The physical – new paragraph).

In introduction it is worth mentioning that not every method is suitable for different cells (plant, animal, bacteria).

Line 98 –DNA quality and quantity (in case of DNA introduction? Or did you mean DNA of host)?)

Materials and Methods

Plasmids  - it should be indicated what the size of plasmid is.

How can we compare the methods if two different plasmids (mVenus C1 and pmaxGFP) are used?

There is no chapter about the assessment of cell viability method (cell viability is mentioned 25 times).

How a confluence of cells was assessed?

2.4.1. – if this protocol is from Lonza, there is no need to write it here down (link).

2.6. GFP expression was tested after 24h, 5d, 7d by flow cytometry.  Why wasn’t a cell sorting done after 24h?

2.7 – it is not clear which target gene was analysed. What does REEP5 name stand for?

Results

Line 317 – cytotoxicity – Did you assess cytotoxicity in this paper? How did you assess cytotoxicy (line 322).

Line 322 – higher number of cells (How did you assess number of cells?)

Figure 4 3A - What does control mean here?

Line 458 – 2x105  were taken as 100% viable – why?

Line 464 – we used 3x105 ….. cells – why?

References

36 references and only 9 newest (2018-2022) – need to be improved!  
